# Effects of Salinity on Physiological, Biochemical and Gene Expression Parameters of Black Tiger Shrimp (*Penaeus monodon*): Potential for Farming in Low-Salinity Environments

**DOI:** 10.3390/biology10121220

**Published:** 2021-11-23

**Authors:** Md. Lifat Rahi, Khairun Naher Azad, Maliha Tabassum, Hasna Hena Irin, Kazi Sabbir Hossain, Dania Aziz, Azam Moshtaghi, David A Hurwood

**Affiliations:** 1Fisheries and Marine Resource Technology Discipline, Life Science School, Khulna University, Khulna 9208, Bangladesh; lifatrahi@gmail.com (M.L.R.); khairunnaherku@gmail.com (K.N.A.); tabassummaliha65@gmail.com (M.T.); hasnahenairin160628@gmail.com (H.H.I.); bdsabbir@hotmail.com (K.S.H.); 2Department of Aquaculture, Faculty of Agriculture, University Putra Malaysia (UPM), Serdang 43400, Malaysia; 3International Institute of Aquaculture and Aquatic Sciences (I-AQUAS), University Putra Malaysia (UPM), Port Dickson 70150, Malaysia; azi.moshtaghi@yahoo.com (A.M.); d.hurwood@qut.edu.au (D.A.H.)

**Keywords:** hemolymph osmolality, stress hormone, gene expression, osmoregulation

## Abstract

**Simple Summary:**

White spot disease is the major obstacle for black tiger shrimp production that cannot spread in freshwater conditions. The present study was conducted to investigate the effects of different salinity levels on the production performance of tiger shrimp. Results indicate that low salinity stress (particularly freshwater) significantly reduces growth rate initially. Following an initial acclimation phase (up to 30 days), tiger shrimp perform regular growth. Overall, results showed farming potential of black tiger shrimp at freshwater environments (with minimal effects on production performance) that can help minimizing outbreaks of white spot disease.

**Abstract:**

Salinity is one of the most important abiotic factors affecting growth, metabolism, immunity and survival of aquatic species in farming environments. As a euryhaline species, the black tiger shrimp (*Penaeus monodon*) can tolerate a wide range of salinity levels and is farmed between brackish to marine water conditions. The current study tested the effects of six different salinity levels (0‰, 2.5‰, 5‰, 10‰, 20‰ and 30‰) on the selected physiological, biochemical and genetic markers (individual changes in the expression pattern of selected candidate genes) in the black tiger shrimp. Experimental salinity levels significantly affected growth and survival performance (*p* < 0.05); the highest levels of growth and survival performance were observed at the control (20‰) salinity. Salinity reductions significantly increased free fatty acid (FFA), but reduced free amino acid (FAA) levels. Lower salinity treatments (0*–*10‰) significantly reduced hemolymph osmolality levels while 30‰ significantly increased osmolality levels. The five different salinity treatments increased the expression of osmoregulatory and hemolymph regulatory genes by 1.2*–*8-fold. In contrast, 1.2*–*1.6-fold lower expression levels were observed at the five salinity treatments for growth (alpha amylase) and immunity (toll-like receptor) genes. O_2_ consumption, glucose and serotonin levels, and expression of osmoregulatory genes showed rapid increase initially with salinity change, followed by reducing trend and stable patterns from the 5th day to the end. Hemocyte counts, expression of growth and immunity related genes showed initial decreasing trends, followed by an increasing trend and finally stability from 20th day to the end. Results indicate the farming potential of *P. monodon* at low salinity environments (possibly at freshwater) by proper acclimation prior to stocking with minimal effects on production performance.

## 1. Introduction

Farming of marine species (e.g., Penaeid shrimps) in freshwater can provide the benefit of eliminating usual pathogenic loads from the natural marine environment [1,2], also providing farming opportunities in inland regions where seawater is difficult to access. For example, white spot disease caused by the white spot syndrome virus (WSSV) is the most widespread and severe disease, potentially causing total mortality in farmed Penaeid shrimp [3]. Previous investigations showed that WSSV cannot readily spread or cause disease outbreak in freshwater conditions [4]. Therefore, growing Penaeid shrimp in freshwater can help to overcome the challenge of disease outbreaks, thereby improving aquaculture production. However, the transition to freshwater can pose a significant threat to organismal growth, immunity, survivability and overall biological processes [5,6]. The ability to minimize the negative effects largely depends on the target species; euryhaline crustacean species (e.g., *Penaeus monodon*) can provide ideal systems to test these effects [7]. In general, euryhaline species are able to rapidly regulate their internal biological processes (physiological, biochemical, cellular and genetic) to cope with the environmental (e.g., salinity) changes.

The main challenge for acclimation and survival in low-salinity environments is to maintain ionic regulation (osmoregulation) between body fluid (hemolymph in crustaceans) and the surrounding medium [8,9,10]. Organisms must regulate internal physiological, biochemical and genetic mechanisms rapidly to deal with salinity changes [2,11,12,13]. Large-scale change in environmental salinity levels can impose severe osmotic stress on organisms that ultimately results in slower growth performance, increasing disease susceptibility and mortality. In this regard, crustaceans are well known for their ability to rapidly regulate these internal biological processes to deal with salinity fluctuations [14,15].

The main physiological aspects of changing osmoregulatory function due to salinity change involve: (i) changes in the ionic content (uptake or release of ions based on external environmental salinity) of body fluid (change in hemolymph osmolality); (ii) changes in gill ultra-structure, extension or contraction of gill lamellae; and (iii) regulatory cell volume increase (RVI) or decrease (RVD), depending on external salinity [13,16,17,18]. Biochemical aspects involve changes in: (i) the amount of free fatty acids (FFA) and free amino acids (FAA) in the hemolymph and in the gill (that create an impermeable membrane to restrict diffusive ion loss or gain) to establish a condition of osmotic equilibrium [19,20]; (ii) the number of hemocyte cells in the hemolymph that indicate immunity status (hemocyte cells break down under stressful conditions and the amount of hemocyte cell lysis depends on the intensity of stress) [20]; and (iii) increased serotonin (crustacean stress hormone) and glucose levels in the hemolymph [3]. Genetic or genomic responses may include changes in expression patterns of certain genes, modifying mRNA production for subsequent protein synthesis [10,21,22,23].

Of these biological mechanisms, genetic response (changes in gene expression patterns) is the most rapid, occurring immediately with salinity fluctuation [12,21], showing increasing trends initially, followed by a decrease from the peak and finally a basal, stable pattern [13,18,21,24]. Physiological and biochemical responses occur at a later stage (occurring slowly and gradually), triggered by the alterations in gene expression pattern [10]. These physiological, biochemical and genetic responses require increased energy expenditure. Therefore, organisms consume more O_2_ to meet the growing demand for energy to respond to the environmental changes. This extra energy is used for osmoregulation (maintaining ionic balance due to salinity change) instead of growth.

The black tiger shrimp (*Penaeus monodon*) is the second most farmed crustacean species globally (after *Litopenaeus vannamei*), with a market value of ≈$US 7.35 billion in 2018 [25,26]. *P. monodon* is a major coastal aquaculture species in South-east Asia where it contributes significantly to the economy of many countries [27,28]. For example, Bangladesh earned ≈$US 300 million by exporting 29,000 metric ton (MT) of *P. monodon*; currently, it is the second-largest export earning source for the country [29,30].

*P. monodon* is a marine (euryhaline) species, but naturally tolerates a wide range of salinity levels (5*–*35‰) [5]. The optimum salinity range for *P. monodon* farming has been shown to be from 10*–*20‰ [7]. In some countries however, *P. monodon* farming has proven successful at lower salinities. In Bangladesh, for example, farming is prevalent at salinities ranging between 5*–*20‰ [31,32]. In recent years, many farmers have been stocking *P. monodon* at even lower salinity levels (2*–*3‰, where giant freshwater prawn (*Macrobrachium rosenbergii*) farming was practiced previously); salinity levels in these farming areas convert to freshwater (0‰) during the monsoon season due to intense rainfall [33]. Moreover, *P. monodon* farming with rice is also practiced as a replacement for *M. rosenbergii*. Farming practice of *P. monodon* in lower salinities (in freshwater) provides slower growth and production performance [2]. Although black tiger shrimp (*P. monodon*) survives and grows at lower salinities (even in freshwater), the severe osmoregulatory stress imposed on individuals results in slower growth and increased mortality [1,20].

While *P. monodon* has shown the ability to survive and grow in low-salinity environments (also in freshwater), no studies have been directed at testing the lower bounds of salinity tolerance (physiological, biochemical and genetic performance), particularly at 0‰ for this species. Several studies have been conducted to test physiological and biochemical changes in *P. monodon* due to salinity changes between 5*–*35‰ [1,20]. However, investigations based solely on physiological or biochemical or genetic aspects cannot provide sufficient insights to decipher the underlying mechanisms involved with acclimation to wider range of salinities.

Previous genomic studies on different aquatic crustaceans revealed 43 candidate genes associated with osmoregulation and ionic balance [18,22,23,34], 26 growth related genes [10,35,36] and 39 genes for disease resistance [37,38,39]. These genomic resources provide ample opportunities to infer different acclimation mechanisms and consequences with regard to environmental manipulations (under specific experimental conditions). Investigations on integrative biological aspects (physiological, biochemical and genetic) from extreme low (0‰) to high (30*–*35‰) environmental salinity levels can provide powerful insight for inferring the farming potential of tiger shrimp at low-salinity environments.

The present study was conducted to investigate the physiological (growth, rate of O_2_ consumption, hemolymph osmolality and hemocyte count), biochemical (FAA, FFA, serotonin and glucose levels in the hemolymph) and genetic changes (expression of eight candidate genes: four osmoregulatory genes, two hemolymph regulatory genes, one growth and one immune response gene) of *P. monodon* at six different experimental salinity levels (0‰, 2.5‰, 5‰, 10‰, 20‰ and 30‰) for a period of 60 days.

## 2. Materials and Methods

### 2.1. Experimental Shrimp Collection

Specific pathogen free (SPF) juvenile individuals (≈0.2 g) of black tiger shrimp from the same cohort were obtained from a commercial hatchery (Panna SPF Hatchery), Khulna, Bangladesh. A total of 1000 juveniles were collected from the hatchery for this study. Shrimp larvae were transported in five different plastic bags (200 individuals in each bag) with oxygen supplied. Salinity in the transportation bags was 20‰ (same salinity as maintained in the hatchery for rearing).

### 2.2. Experimental Tank Setting and Acclimation

Prior to sample collection, 18 (30 L) glass tanks were prepared for rearing experimental shrimp under six different salinity levels (three replicated tanks for each salinity). Experimental tanks were filled with 20‰ water (up to 25 L) and maintained with continuous aeration. Juvenile shrimp were brought to the Water Quality Laboratory of Fisheries and Marine Resource Technology (FMRT) Discipline, Khulna University and randomly allocated to tanks (50 individuals per tank). Juveniles were acclimated in the experimental tanks for 10 days. Experimental shrimps were fed with a commercial feed (Sano S-PAK, CP Feed, Bangkok, Thailand) twice per day (protein content of the feed 40%) at the rate of 10% of the total biomass.

### 2.3. Salinity Stress Experiment

Following 10 days of acclimation to the tank environment, we started the salinity stress experiment. The target salinity levels were 0‰, 2.5‰, 5‰, 10‰, 20‰ (as control) and 30‰. Salinity levels were increased or decreased by 2.5‰ per day to achieve the target salinity levels. Salinity was reduced in two phases at six hour intervals (reduce 1.5‰, wait for 6 h and then reduction of 1‰) to impose minimum stress on shrimp during the salinity transfer stage. Salinity levels were increased by mixing appropriate amounts of brine water (150‰) while salinity levels were decreased by adding freshwater to the tanks. For 0‰, 2.5‰ and 5‰ treatments, salinity reductions were started 2*–*4 days before 10‰ and 30‰ treatments to achieve all five treatment salinity levels simultaneously.

### 2.4. Growth and Survival Performance

Experimental shrimp were sampled at 15 day intervals (30 individuals per sampling time) to measure body weight. Survival rates were determined by counting the number of individuals at the end of the experiment. The following equations were used to measure specific growth parameters [40]:
DWG (%) = ((BW_f_ − BW_i_)/(BW_i_ × t) × 100(1)
SGR (%) = ((ln BW_f_ − ln BW_i_)/t) × 100(2)
FCR = feed intake (g)/weight gain (g)(3)
where, DWG = daily weight gain, BW_f_ = final body weight, BW_i_ = initial body weight, t = total experimental time (60 days), SGR = specific growth rate, and FCR = feed conversion ratio.

### 2.5. Rates of Oxygen Consumption

Salinity induced changes in the rate of O_2_ consumption were measured according to [10] and [41]. In brief, individual shrimp were transferred to 250 mL flow respirometric chambers (Q-Box Aqua Respiratory System, Qubit, Ontario, Canada). Rate of water flow (FR) in the respirometric chamber was 1.5 Lh^−1^. O_2_ consumption rates were estimated by subtracting O_2_ exit concentration (O_2_ ex) from the concentration at entry (O_2_ en) in each chamber using the following equation [41]:O_2_ consumption = [O_2_ en − O_2_ ex] × FR(4)

O_2_ consumption levels were then adjusted with respect to the control chamber levels (without any shrimp). Shrimp individuals were weighed immediately after the measurement of O_2_ consumption to estimate the O_2_ consumption rates (mg O_2_ h^−1^ g^−1^). Five replicate shrimp were used for measuring the rates of O_2_ consumption for each salinity level. Samples were collected from 10 different time intervals (0 H, 12 H, 24 H, Day 2, Day 3, Day 4, Day 5, Day 15, Day 30 and Day 60) for measuring O_2_ consumption rates, where 0 H represents sampling immediately after achieving the respective salinity levels.

### 2.6. Assaying Hemolymph Osmolality

Hemolymph samples were collected from three *P. monodon* individuals at each sampling time. 100 µL of hemolymph was collected using a 500 µL (hypodermic 23-gauge needle) syringe pre-filled with 100 µL of anticoagulant (0.1% glutaraldehyde in 0.2 M sodium cacodylate, pH 7.0). Hemolymph was collected by inserting the needle through the inter-segmental membrane between the cephalothorax and the first abdominal segment. Half of the anticoagulated hemolymph was used for measuring osmolality while the remaining half was used for hemocyte counts. For measuring hemolymph osmolality, another 50 µL of anticoagulant was added in the 100 µL mixture (50 µL hemolymph + 50 µL anticoagulant) to double the volume of anticoagulant to hemolymph (50 µL hemolymph + 100 µL anticoagulant). Hemolymph osmolality of the mixture was then measured using a cryoscopic osmometer (Osmomat 030, Gonotec, Berlin, Germany). Three replicates were used for hemolymph collection. Osmolality of the anticoagulant was also measured to precisely calculate the salinity induced changes in hemolymph osmolality by applying the following equation [42]:Hemolymph osmolality = 3 × osmolality of sample mixture − 2 × osmolality of anticoagulant

Hemolymph osmolality was also measured at 0 H, 12 H, 24 H, Day 2, Day 3, Day 4, Day 5, Day 15, Day 30 and Day 60.

### 2.7. Total Hemocyte Counts

Hemocytes were counted according to the methods outlined in [43]. In brief, 100 µL of anticoagulated hemolymph was fixed with an equal volume of neutral buffered formalin (10%) for 30 min to measure the total hemocyte count (THC). Fixed samples (200 µL) were serially diluted 2, 4, 8, 16, and 32 times using ice-cold phosphate buffered saline (PBS, 20 mM, pH 7.2). Total hemocytes were then counted by using a hemocytometer (Boeco, Hamburg, Germany) and counting the number of cells under a microscope (SOLARIS-T-LED, Rome, Italy) at 100× magnification. Total hemocyte counts were then expressed as the number of cells per ml of hemolymph.

### 2.8. Assaying Glucose and Serotonin Levels in the Hemolymph

Hemolymph glucose levels were determined using the glucose oxidase method according to [44]. For this step, fresh hemolymph (50 µL) was collected from experimental individuals (collected separately from the same shrimps during hemolymph collection in the previous steps). Freshly collected hemolymph samples were centrifuged at 1000 rpm for 10 min at room temperature to obtain cell free hemolymph (CFH) by removing the liquid (supernatant). 1 mL of glucose mono reagent (Span Diagnostics Ltd., Delhi, India) was added to 10 µL CFH and incubated at room temperature for 10 min. Finally, 200 µL of the sample was added to the well plates of ELISA reader (Cyberlab Inc., Los Angeles, CA, USA) to read the glucose levels at 492 nm absorbance. In addition, 100 µL of freshly collected hemolymph was used for assaying serotonin levels using a serotonin creatinine sulfate standard and the serotonin primary antibody according to [45].

### 2.9. Free Amino Acid (FAA) Levels in Hemolymph

FAA levels in the hemolymph of individual experimental shrimp were determined according to the methods outlined by [46]. In brief, fresh hemolymph samples were deproteinized by adding an equal volume (50 µL) of 12% trichloroacetic acid and mixed well using a vortex mixer. Samples were then placed in a freezer at 4 °C for 20 min, following which the mixture was centrifuged at 10,000× *g* (4 °C) for 10 min. The supernatants were collected and 6M NaOH solution was added to the supernatant to adjust the pH level to 2.2. Finally, FAA analysis was performed using a Hitachi L-8900 amino acid analyzer (Hitachi, Tokyo, Japan) with a Li column.

### 2.10. Free Fatty Acid (FFA) Levels

Gill tissues were dissected from the experimental shrimp for FFA analysis. Dissected gills were then homogenized and freeze-dried, individually. Total lipids were extracted using the chloroform-methanol (2:1, *v*/*v*) method according to [47]. Total lipids were then esterified with 14% boiling boron trifluoride/methanol (*w*/*w*) which was subsequently used for fatty acid methyl esters (FAMEs) extraction by using hexane. FAME samples were injected into a Thermo Trace GC Ultra gas chromatograph (100 m × 0.25 mm ID, 0.2 μm film thickness) fitted Supelco SP-2560 capillary column (Supelco, Bellefonte, PA, USA) for analysis using flame ionization detection (FID) according to the methods outlined in [46]. Following this step, the peaks for fatty acids were identified by comparing retention times with a known standard (Sigma-Aldrich Co., St. Louis, MO, USA). Finally, fatty acid profiles were expressed as percentage of each fatty acid to the total fatty acids (% total fatty acids).

### 2.11. Gene Expression Study

Different tissues of black tiger shrimp were used for investigating expression pattern of different gene categories. Gill tissue was used for investigating expression of osmoregulatory genes, while hepatopancreas was used for the expression of growth and immune response genes. Samples were collected at the 10 different time intervals identified earlier, where 0 H represents sampling immediately after salinity shock. Three shrimp were dissected at each sampling time from each salinity level for the gene expression study. Freshly dissected gill and hepatopancreas tissues were used for total RNA extraction using the TRIzol/chloroform extraction method followed by RNA purification using an ISOLATE II RNA Mini Kit (Cat # 52072, Bioline, UK) according to the manufacturer’s protocol. Quality and quantity of the extracted RNA were checked using 2% agarose gel electrophoresis and a NanoDrop 2000 Spectrophotometer (Thermo Scientific, Massachusetts, USA). RNA samples were preserved at −80 °C for subsequent use. Complementary DNA (cDNA) synthesis was performed (using 1 µg of total RNA for each sample) by using a SensiFAST cDNA synthesis kit (Cat # BIO-65054, Bioline, UK) according to the manufacturer’s protocol. cDNA samples were then preserved at −20 °C for further analysis.

Following a comprehensive literature review on crustaceans, candidate genes were selected based on their inferred functional roles [7,22,23]. Primer sequences of the eight candidate genes for *Penaeus monodon* were obtained (Table 1) from earlier studies [7,48,49]. EF1α was used as the reference gene because it was previously found to be a suitable reference gene for RT-qPCR studies in *P. monodon* [49,50,51]. The eight candidate genes were under four different functional categories including; (i) two hemolymph regulatory genes: crustacean hyperglycemic hormone (CHH) and diuretic hormone (DH); (ii) four osmoregulatory genes: Na^+^/K^+^-ATPase (NKA), Na^+^/H^+^ exchanger (NHE), Na^+^/K^+^/2Cl^−^ co-transporter (NKCC) and V-type H^+^ ATPase (VTA); (iii) one growth related (or digestive) enzyme (α-Amylase) and iv) one immune response gene: Toll-like receptor (TLR).

For the RT-qPCR assay, reactions were performed on an RG-600 thermal cycler (Corbett, Australia) using a SensiFAST SYBR No-ROX kit (Bioline, London, UK) according to the manufacturer’s protocol. Reaction conditions for qPCR included several steps: (i) polymerase activation (95 °C for 2 min) and (ii) 40 cycles of: denaturation (95 °C for 5 min), annealing (54*–*60 °C for 20 s) and a final extension at 72 °C for 20 s [10,12,52]. Three technical replicates were used for each sample. Standard melt-curve analyses were performed following each reaction to ensure the amplification of a single qPCR product. Relative gene expression values were then estimated using the ΔΔC_T_ method [53]. Relative gene expression values were used for testing statistical significance in the SPSS software package (version 23). Gene expression values were also loaded in the R statistical package for preparing graphs and also for principal components analysis (PCA) [54].

### 2.12. Data Analysis

SPSS 23 software was used to test level of significance (α = 0.05) using two-way ANOVA (followed by Duncan’s multiple range test), to assess the effects of different salinity levels and sampling times on the physiological (growth, O_2_ consumption, hemolymph osmolality and hemocyte counts), biochemical (hemolymph glucose, FAA and FFA) and genetic (changes in gene expression pattern) responses of *Penaeus monodon*. Finally, regression analyses were performed between different biological markers, also using SPSS 23.

## 3. Results

### 3.1. Growth and Survival Performance of Black Tiger Shrimp at Different Salinity Levels

Experimental salinity levels significantly altered growth performance of individual shrimp over times (F_5,125_ = 86.3, *p* = 0.03). The highest growth performance was obtained at the control salinity (20‰), while the lowest was at 0‰ (Table 2 and Figure 1). Figure 1 also shows a decreasing trend in growth rate with salinity reduction from the control. At 20‰ (control), significantly higher growth (*p* < 0.05) was observed over all other salinity treatments for the entire experimental period. No significant differences were observed between 10‰ and 30‰ up to 45th day but by the end (60th day), significantly higher growth was obtained at 30‰. Initially (up to 15th day), no significant differences were obtained between 2.5‰ and 5‰ but significantly higher growth was achieved at 5‰ from 30th day to the end of this experiment (Figure 1). Experimental salinity levels also impacted survivability of *Penaeus monodon* individuals (Table 2). Consistent growth rate was obtained at the control salinity (20‰) throughout while very lower growth rates were obtained at low salinity levels (0*–*5‰) up to 30th day, following which growth rates improved (Appendix A).

### 3.2. Changes in O_2_ Consumption Rates

Experimental salinity changes significantly (*p* < 0.05) altered O_2_ consumption rates (0.66*–*1.48 mg O_2_ h^−1^ g^−1^ depending on experimental salinity levels) in *P. monodon* (Figure 2). No significant differences were observed between 20‰ and 30‰ salinities from the beginning to the end of the experiment. In the 0*–*10‰ treatments, O_2_ consumption rates increased with reduced salinity levels (10‰ < 5‰ < 2.5‰ < 0‰) (Figure 2). Significantly higher O_2_ consumption rates (*p* < 0.05) were observed at 0‰ throughout the experiment compared to the other salinity levels. The lower salinity levels (0‰, 2.5‰ and 5‰) showed 1.8*–*2.5 fold higher O_2_ consumption rates than higher salinity levels (10‰, 20‰ and 30‰). Initially (at 0 H), no significant differences were observed between the three higher salinity levels, following which (from 12 H to the end) 10‰ showed significantly higher (*p* < 0.05) O_2_ consumption rates over 20‰ (control) and 30‰.

### 3.3. Hemolymph Osmolality Changes

Hemolymph osmolality levels of individual black tiger shrimp were affected by both the salinity and sampling times from 12 H onward (Figure 3). No significant differences were observed initially (0 H) between salinity levels. The highest osmolality levels (698–760 mOsm/kg H_2_O) were observed at 30‰ (*p* < 0.05) from 24 H to the end of this study. Both at 20‰ and 30‰ salinities, significantly higher osmolality levels were observed throughout the experiment (from 12 H to the end) compared to the four lower salinities. A gradual increase in osmolality was observed at 30‰ from 24 H to the 4th day, following which it was stable for the remainder (Figure 3). No significant differences were observed between 0‰, 2.5‰, 5‰ and 10‰ up to the 2nd day (635*–*670 mOsm/kg H_2_O depending on salinity levels). Significantly higher osmolality levels (*p* < 0.05) were observed at 5‰ (645*–*655 mOsm/kg H_2_O) and 10‰ (653*–*665 mOsm/kg H_2_O) compared to 0‰ (600*–*610 mOsm/kg H_2_O) and 2.5‰ (603*–*615 mOsm/kg H_2_O) from the 3rd day to the end of this experiment (but no significant differences were observed between 5‰ and 10‰). Significant differences between 0‰ and 2.5‰ were observed only from the 20th day to the end of the experiment (Figure 3).

### 3.4. Total Hemocytes in the Hemolymph

Salinity treatments significantly altered total hemocyte counts (30*–*45 million hemocyte cells/mL) in the hemolymph of experimental shrimp (Figure 4). Salinity reductions (0‰, 2.5‰, 5‰ and 10‰) from the control (20‰) reduced hemocyte counts by 5*–*25%. Salinity increase (30‰) also reduced hemocyte counts (37*–*39 million cells/mL) significantly up to the 4th day, following which no significant differences were observed between 20‰ (control) and 30‰. Significantly higher hemocyte counts (*p* < 0.05) were observed at control salinity (20‰) than at 30‰ from 12 H to the 4th day (40*–*43 million cells/mL), following which no significant differences were observed between these two salinities. While 20‰ and 30‰ showed significant differences over 10‰ up to the 5th day, no significant differences were observed between these three salinities from the 10th day to the end of this study. Different low salinity treatments reduced hemocyte counts rapidly up to the 3rd day (30*–*36 million cells/mL), then showed gradual increase up to the 30th day (38*–*43 million cells/mL) and finally stability until the end of the experiment.

### 3.5. Hemolymph Glucose and Serotonin Levels

Glucose and serotonin levels in the hemolymph showed trends similar with salinity changes (Figure 5 and Figure 6). Both glucose and serotonin levels increased rapidly up to the 2nd day, followed by gradual decrease up to the 10th day and finally stability towards the end. Levels of hemolymph glucose and serotonin in the salinity treatments were ranked as 0‰ (51*–*94 µg/mL and 52*–*99 ng/mL) > 2.5‰ (50*–*87 µg/mL and 50*–*93 ng/mL) > 5‰ (50*–*84 µg/mL and 50*–*80 ng/mL) > 10‰ (49*–*78 µg/mL and 49*–*75 ng/mL) > 30‰ (50*–*74 µg/mL and 48*–*72 ng/mL), while these levels were stable (50*–*54 µg/mL and 49*–*52 ng/mL) at control salinity (20‰) throughout the experiment. All five salinity treatments (0‰, 2.5‰, 5‰, 10‰ and 30‰) showed significantly higher (*p* < 0.05) glucose and serotonin levels over the control (20‰) salinity.

### 3.6. Salinity Induced Changes in Free Amino Acids (FAAs) and Free Fatty Acids (FFAs)

Amino acid profiling of shrimp hemolymph revealed 27 different amino acids (including essential amino acids) and amino acid derivatives (Table 3). Total essential amino acids (∑EAAs) showed general declining trends with salinity reductions. The highest levels of EAAs were observed at 30‰ (759.5 nmol mL^−1^); significantly higher EAAs were observed (*p* < 0.05) at 20‰ and 30‰ compared to the other test salinities. Total free amino acids (TFAAs) showed similar trends, with the highest levels observed at 20‰ (3708.6 5 nmol mL^−1^), but no significant differences between 20‰ and 30‰. Three particular amino acids (alanine, glycine and proline) showed the highest levels at 0‰ salinity. Profiling of fatty acids in the gill tissue showed increasing trends with decreasing salinity levels (Table 4). No significant differences were observed between 0‰, 2.5‰ and 5‰ salinities for different types of fatty acids, while these three salinity treatments showed significantly higher (*p* < 0.05) levels of fatty acids compared to the control (20‰).

### 3.7. Changes in Gene Expression

Genes in different categories were affected differently by the experimental salinity levels (Figure 7, Figure 8, Figure 9, Figure 10, Figure 11, Figure 12, Figure 13 and Figure 14). The four ion regulatory genes; Na^+^/K^+^-ATPase (NKA), Na^+^/K^+^/2Cl^−^ Co-transporter (NKCC), Na^+^/H^+^ exchanger (NHE) and V-type H^+^ ATPase (VTA); showed salinity specific differential expression levels across the sampling times. At 0‰, NKA showed 1.5*–*3 fold higher (Figure 7), NHE showed 4*–*6 fold higher (Figure 8) and VTA showed 3*–*8 fold higher (Figure 9) expression compared to the other test salinities. These three genes showed generally increasing levels of expression with decreasing salinity. NKCC showed a reverse pattern (Figure 10), a general increase in expression with elevated salinity levels, with 1.5*–*4 fold higher expression levels at 30‰ compared to other test salinities. All four ion regulatory genes (NKA, NKCC, NHE and VTA) showed immediate changes in expression levels and became stable within 4*–*5 days with increasing and decreasing expression through to the 4th day.

Crustacean hyperglycemic hormone (CHH) and diuretic hormone (DH) showed contrasting patterns in expression with salinity changes (Figure 11 and Figure 12). The highest levels of CHH expression were observed at 30‰ (1.1*–*2 fold higher than at the other salinity levels), while the highest DH expression was obtained at 0‰ (1.2*–*2.5 fold higher). Both of these genes showed lowest expression levels at control (20‰) salinity. CHH showed gradual increase at 2.5‰, 5‰ and 10‰ up to the 2nd day (peak expression levels), following which expression declined up to the 5th day and then remained stable for the remaining time. At 0‰, increasing CHH expression continued up to the 3rd day, following which it declined to the 20th day, and finally was stable towards the end. DH showed sharply increasing trends in expression up to the 4th day (peak levels), following which gradual declining to the 30th day and finally a stable at 0‰.

Consistently higher expression of the enzyme (α-Amylase) and immune response gene (Toll like receptor) (Figure 13 and Figure 14) were observed at the control salinity (20‰) from the beginning to the end of this study. For all five salinity treatments (0‰, 2.5‰, 5‰, 10‰ and 30‰), gradually decreasing α-Amylase expression was observed up to the 5th day, followed by increase to the 30th day and finally stable expression towards the end (Figure 13). TLR showed similar trends in expression (Figure 14). Significant differences were observed for TLR expression among experimental salinities from beginning to the end. Both α-Amylase and TLR showed 1.2*–*2 folds higher expression levels at control (20‰) compared to the five salinity treatments.

## 4. Discussion

Salinity treatments significantly altered physiological (growth, survival, O_2_ consumption and hemolymph osmolality), biochemical (FAAs, FFAs, hemocyte counts, glucose and serotonin levels) and genetic (expression pattern of selected candidate genes) parameters of black tiger shrimp (*Penaeus monodon*). Any change in environmental salinity levels imposes osmotic stress on aquatic organisms. Stress levels depend on the intensity of salinity change due to differences in osmolality between organismal body fluid (hemolymph in crustaceans) and the surrounding environment [6,11,57]. At high salinity environments, crustaceans can act as osmo-conformers, maintaining hemolymph osmolality below the aquatic medium while they act as osmo-regulators in low salinity environments [17,23,58]. This osmotic stress, in turn, can affect overall organismal biology of black tiger shrimp including, physiology (growth, metabolic performance and immunity), biochemistry (hormonal imbalance, changes in hematological parameters, and FAAs and FFAs) and gene expression. These diverse changes will retard growth, increase mortality, increase susceptibility to diseases and will adversely affect overall wellbeing. As such, we observed reduced growth and survival with salinity changes from the control salinity (20‰) (Figure 1 and Table 2).

Low salinity treatments (0*–*5‰) significantly (*p* < 0.05) retarded growth of black tiger shrimp (Figure 1), while it increased the rates of O_2_ consumption (Figure 2). Higher O_2_ consumption indicates increase of metabolic rate that potentially leads to the osmotic stress response mechanism to salinity change [59,60]. Under any stressful condition (i. e., salinity shock), the energy requirement of aquatic organisms is elevated immediately to counterbalance the adverse effects of stressors. Organisms typically meet the growing energy demand by consuming more food (with consequent increase in metabolic rates) or by using the body’s reserve energy sources that ultimately reduce growth, immunity and survival [10,11,15,61]. The highest rates of feed intake (Table 2) and O_2_ consumption (Figure 2) coupled with the lowest growth (Figure 1) at 0*–*5‰ treatments indicate response to osmotic stress by the experimental black tiger shrimps at the expense of growth retardation. The lowest FCR at 20‰ (0.89) coupled with the highest growth indicate no imposed osmotic stress at this control salinity; providing efficient use of feed for optimal growth.

Although 20‰ was the control salinity (larvae reared in the hatchery and also in the laboratory at this salinity level), rapid increase in hemolymph osmolality from 24 H to the 3rd day at 30‰ salinity (Figure 3) indicate similar environmental salinity where black tiger shrimp evolved and are naturally distributed. No significant differences in hemolymph osmolality between 5‰ and 10‰, with significant differences between 10*–*30‰ salinities indicate that a strong osmotic gradient is established at every 10‰ salinity change. Crustaceans are well known for their ability to efficiently regulate hemolymph osmolality with salinity changes which enable them to rapidly acclimate to the surrounding environment [6,11,62].

In the control (20‰), shrimp maintained hemolymph osmolality levels between 670*–*686 mOsm/kg H_2_O, which is slightly above the surrounding water (osmolality of 20‰ water = 600 mOsm/kg H_2_O), reflecting a weaker osmotic gradient between hemolymph and 20‰ water which imposed lower or almost no stress. At 30‰ (osmolality = 750 mOsm/kg H_2_O), hemolymph osmolality of experimental shrimps reached 730*–*760 mOsm/kg H_2_O from the 2nd day to the end (60th day), nearly an iso-osmotic condition that enabled them to grow faster (compared to the 0*–*10‰ salinities). A slower and gradual decline in osmolality levels with salinity reduction up to the 3rd day (stable osmolality levels were observed from the 4th day to the end) indicate appropriate physiological adjustments in which salinity change has occurred in experimental shrimps within this timeframe (although there is a strong difference in the ionic/osmotic gradient between hemolymph and low salinities). Moreover, the ability to change hemolymph osmolality levels within a short time indicates the capacity to regularly disperse between brackish and marine environments, and also to adapt to tidally driven changes in salinity at any one location.

In low salinity environments, crustaceans tend to lose ions from body fluid via the gills; increased levels of FFAs in the gill region help to minimize ion loss by creating an impermeable membrane [5,9,19,20,46]. Thus, increasing FFAs in the gills of *P. monodon* with decreasing salinity (Table 4) indicate the response for acclimation (a compensatory mechanism) to salinity reductions. FAAs are also important components of crustacean hemolymph that help to regulate osmolality besides ions [19,20]. Moreover, amino acids can break down immediately after exposure to salinity fluctuations to provide energy sufficient for stress mitigation [5,27]. Therefore, reduced levels of FAAs were observed at 0‰ (and also in other low salinity conditions: 2.5‰ and 5‰) (Table 3) for low salinity specific osmotic stress response as well as for reducing osmolality levels in hemolymph. Lower FAA levels in crustacean hemolymph also inversely affect growth performance [10,19,20]. As such, we observed reduced growth of experimental black tiger shrimps coupled with lower levels of hemolymph FAAs.

Investigating hemocyte counts provides a reliable approach to infer immunity status of organisms, with higher levels of hemocytes potentially indicating better immunity status and reduced counts indicate emerging susceptibility to diseases or pathogens [4,20,63]. Slight variations in hemocyte counts of shrimps in control group (at 20‰) across different sampling times could probably be ascribed to the changes in developmental stages (coupled with faster growth compared to the salinity treatments). Lower hemocyte counts at 0‰, 2.5‰ and 5‰ salinities compared to higher salinities (Figure 4) indicate reduced immunity, most likely due to salinity mediated induced stress. No significant difference for hemocyte counts between 10‰, 20‰ and 30‰ indicates exposure to similar salinity levels where black tiger shrimp are naturally distributed. Although *P. monodon* is a marine species, individuals regularly disperse between the deep sea and inshore coastal waters at different developmental stages of life that enable them to tolerate wide ranging salinity variations without adverse effects on them [26,28,60,64,65]. Results of this study imply that exposure of *P. monodon* to lower salinity environments potentially reduces immunity that will increase susceptibility to different pathogens and diseases. Therefore, special care must be taken (maintaining optimum water quality and feeding) for farming black tiger shrimp in low-salinity environments.

Measuring hemolymph glucose and serotonin (stress hormone) levels in crustaceans indicates the magnitude of stress on experimental individuals [2,45]. Both glucose and serotonin levels are known to negatively affect crustacean growth [3,45,66,67]. Strong positive and significant correlation (Appendix A) between hemolymph glucose and serotonin levels (R^2^ = 0.97, *p* < 0.05) indicate that these parameters respond to stressors in combination simultaneously, and also indicate the intensity of stress experienced by the shrimp at different salinity levels. Higher glucose and serotonin levels at low salinities (0*–*5‰) throughout the course of this experiment imply persistent stress on experimental *P. monodon* individuals; this severe osmotic stress in turn retarded growth performance and survival. A slight decline in serotonin and glucose levels in low-salinity treatments with the progression of the experiment indicates an attempt to reduce or counterbalance osmotic stress levels via acclimation to these low-salinity conditions (0*–*5‰). As such, growth performance at these low salinity levels were found to be gradually increasing (Figure 1). Significantly negative correlations (Figure 15) between growth and serotonin (R^2^ = −0.51, *p* < 0.05) as well as growth and glucose (R^2^ = −0.50, *p* < 0.05) levels further support the interpretation that glucose and serotonin levels indicate the magnitude of stress on experimental shrimp, and that imposed stress impedes growth.

NKA is well known for its important functional role in ionic balance under a wide range of salinities by establishing strong electrochemical gradients [10,11,12,68,69,70]. Very high expression levels (1.5*–*3 fold higher than the control) of NKA at low salinities (0*–*5‰), 30‰ and moderate expression levels (1.2*–*1.5 fold higher) at 10‰ (Figure 7) suggest its important role in ionic regulation under a wide range of salinity levels for black tiger shrimp. VTA and NHE are important osmoregulatory genes, involved with ion absorption in low ionic conditions, and usually exhibit much higher expression levels in freshwater environments [6,13,15,16,70,71]. At 0‰, 4*–*6 fold higher expression of NHE (Figure 8) and 3*–*8 fold higher expression of VTA (Figure 9) compared to the other salinities also reflects the critical role of the products of these two genes in absorbing monovalent ions in freshwater for *P. monodon*. The inferred functional role of NKCC involves releasing salt at high ionic conditions [22,23,34,70]. Approximately 1.5*–*4 fold higher expression levels of NKCC at 30‰ compared to the remaining five salinities (Figure 10) suggest important functional roles of this gene at high ionic conditions (reverse role compared to NHE and VTA). CHH and DH are known to play important functional roles in body fluid regulation in different crustacean species [3,16,72]. A significantly positive correlation between CHH and hemolymph osmolality (R^2^ = 0.46, *p* < 0.05) and a significantly negative correlation between DH and osmolality (R^2^ = −0.45, *p* < 0.05) (Appendix A) indicate salinity specific important functional roles of these two genes (CHH plays a vital role in high-salinity environments, but DH is more important at low-salinity conditions) in body fluid regulation for the black tiger shrimp.

α-Amylase is an important digestive enzyme in crustaceans, showing higher levels of expression in fast growing crustaceans including black tiger shrimp [16,35,36,56,60]. In the current study, significantly higher expression of α-amylase at control salinity (20‰) likely contributed to superior growth performance (Figure 13). TLR acts as a first line of defense against pathogens in crustaceans by recognizing a wide range of pathogen associated molecules and also by triggering the innate immune system [26,73]. Higher expression of TLR at the control salinity (20‰) indicates better immunity status of the tiger shrimps at this stable salinity while reduced expression at the salinity treatments (Figure 14) suggests a declining level of immunity in these shrimp. The compensatory mechanisms to salinity stress by the experimental tiger shrimps likely involved reduced expression of growth and immune response genes, with a consequent increase in the expression of osmoregulatory genes.

Earlier studies implied that crustaceans can acclimate well to salinity changes within 4 to 5 days based on hemolymph osmolality and expression pattern of ion regulatory genes [6,7,10,11,13,57,74,75]. Stable levels of O_2_ consumption, hemolymph osmolality and expression of osmoregulatory genes (NKA, NKCC, NHE and VTA) within five days in this study (at low-salinity treatments) also support these findings. In contrast, it required 20*–*30 days (at 0*–*5‰) to obtain stable levels of glucose, serotonin and hemocyte counts, depending on salinity levels. Moreover, growth rate (fortnightly weight gain) of shrimps at low-salinity treatments (0*–*5‰) was very slow up to the 30th day, while higher growth rate was obtained from the 45th day to the end (Appendix A). These results indicate that *P. monodon* can counterbalance osmotic stress within 4*–*5 days for some of the physiological and genetic (particularly expression of osmoregulatory genes) adjustments with salinity change, while it can take at least 20 days for recovery of overall biological activities including growth, hematological parameters and immunity among others. Thus, we infer that the ability to rapidly regulate different biological mechanisms enable *P. monodon* to acclimate well with large spectrum salinity changes (from freshwater to full strength sea water).

## 5. Conclusions

The present study successfully investigated salinity specific differential changes in some selected physiological, biochemical and genetic responses in the black tiger shrimp (*Penaeus monodon*) to infer the farming potential of this species in low salinity environments. Findings of this study support that with proper acclimation at an early stage in the hatchery, *P. monodon* can be farmed at low ionic and even in freshwater conditions with minimal effect on production performance. In the future, production has to be assessed up to the harvesting stage to validate the farming potential of tiger shrimp in freshwater conditions. Acclimation to low salinity can impose persistent stress on the black tiger shrimp individuals that ultimately will affect immunity (organisms will face the challenge of increased susceptibility to different diseases). Good farming practice (including high quality balanced diet and adequate hygiene) must be maintained to help minimizing outbreaks of disease at low-salinity environments.

## Figures and Tables

**Figure 1 biology-10-01220-f001:**
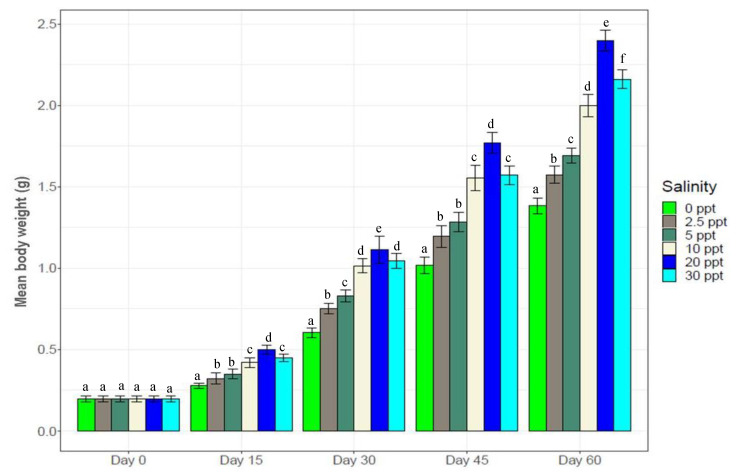
Mean body weight of the black tiger shrimp (*Penaeus monodon*) at every 15 days interval (mean ± S.D.). N = 30 individuals per sampling time. Different letters (a–e) above the bars indicate significant differences among treatments at 5% level of significance.

**Figure 2 biology-10-01220-f002:**
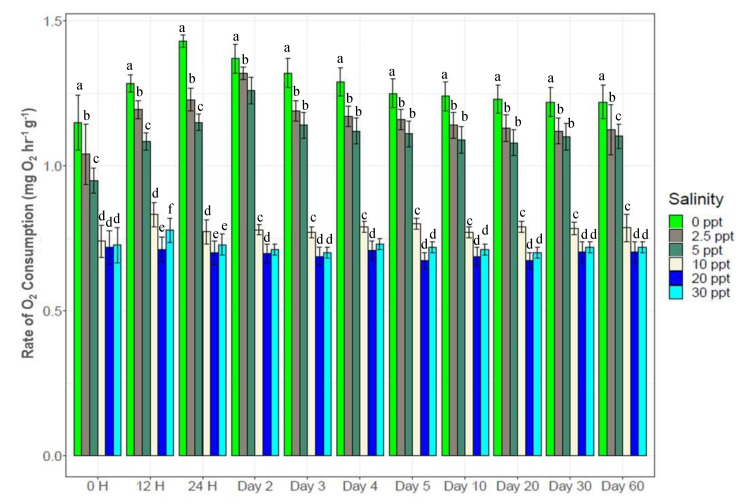
Salinity induced changes in the rate of O_2_ consumption by *P. monodon* (mean ± S.D.). N = 5 individuals per sampling time. Different letters above the bars indicate significant differences at 5% level of significance.

**Figure 3 biology-10-01220-f003:**
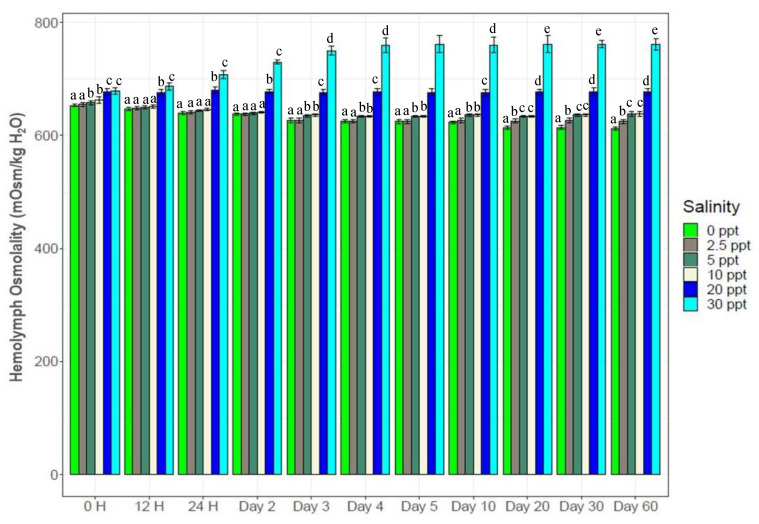
Changes in hemolymph osmolality with salinity change. N = 3 individuals per sampling time. Different letters above the bars indicate significant difference (at 5% level of significance) among the salinity treatments.

**Figure 4 biology-10-01220-f004:**
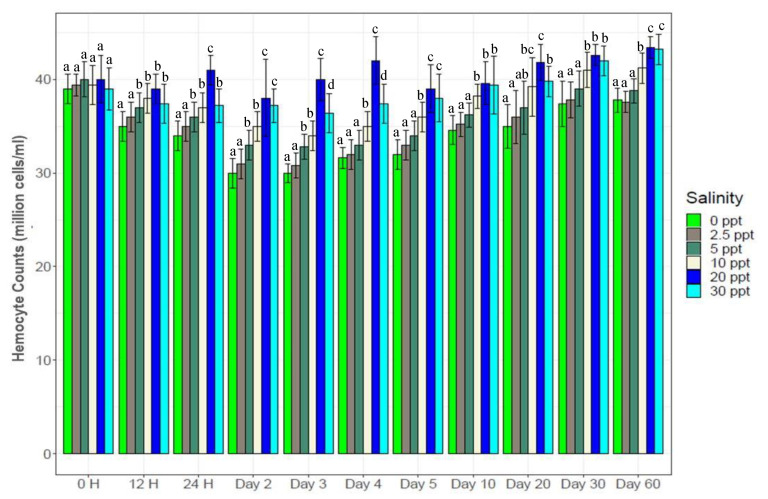
Total hemocyte counts (mean ± S.D.) of black tiger shrimp at six different salinity levels. N = 3 individuals per sampling time. Different letters above the bars indicate significant difference at 5% level of significance among the salinity treatments.

**Figure 5 biology-10-01220-f005:**
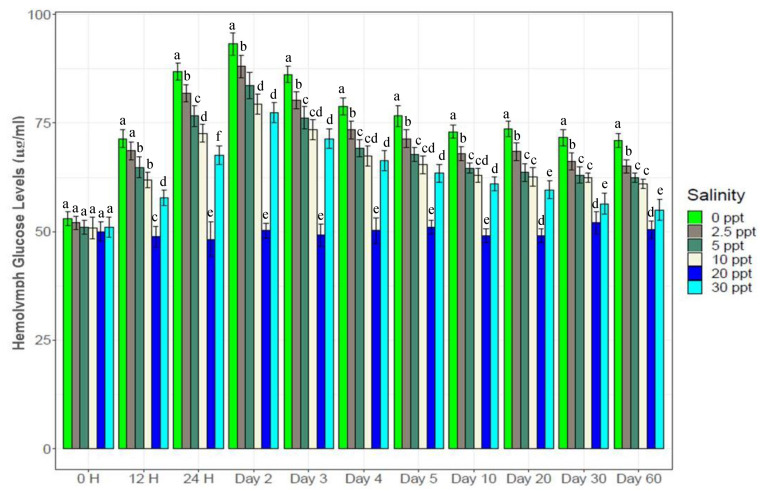
Changes in hemolymph glucose levels (mean ± S.D.) at six different experimental salinities. N = 3 individuals per sampling time. Different letters above the bars indicate significant differences among treatments at 5% level of significance.

**Figure 6 biology-10-01220-f006:**
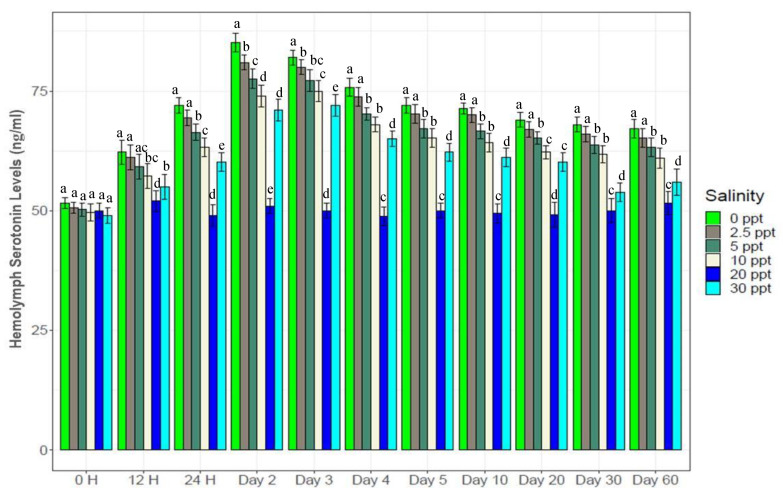
Hemolymph serotinin levels (mean ± S.D.) of black tiger shrimp at the experimental salinities. N = 3 individuals per sampling time. Different letters above the bars indicate significant differences among treatments at 5% level of significance.

**Figure 7 biology-10-01220-f007:**
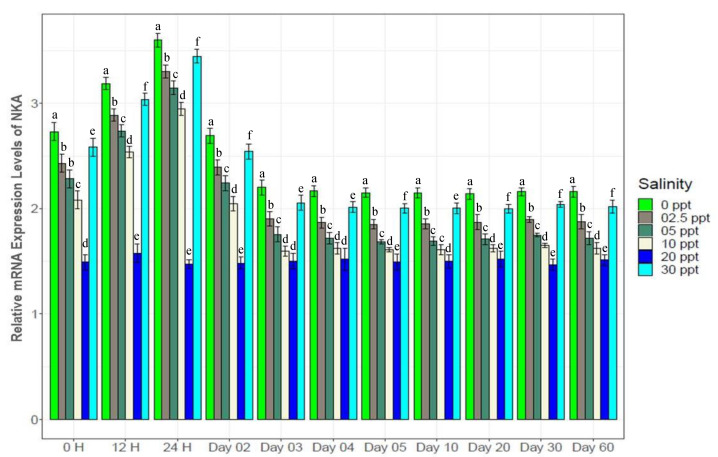
Changes in expression pattern of Na^+^/K^+^-ATPase (mean ± S.D.; N = 3) at six different salinity levels. Different letters above the bars indicate significant differences among treatments at 5% level of significance.

**Figure 8 biology-10-01220-f008:**
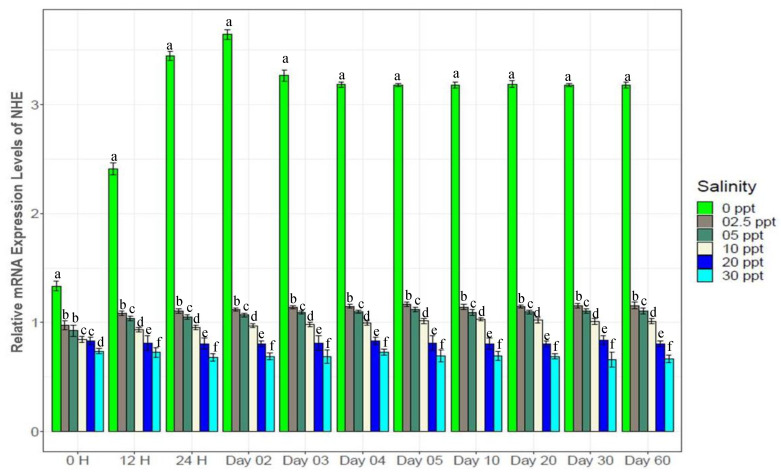
Changes in expression pattern of Na^+^/H^+^-Exchanger (mean ± S.D.; N = 3) at experimental salinities. Different letters above the bars indicate significant differences among salinities.

**Figure 9 biology-10-01220-f009:**
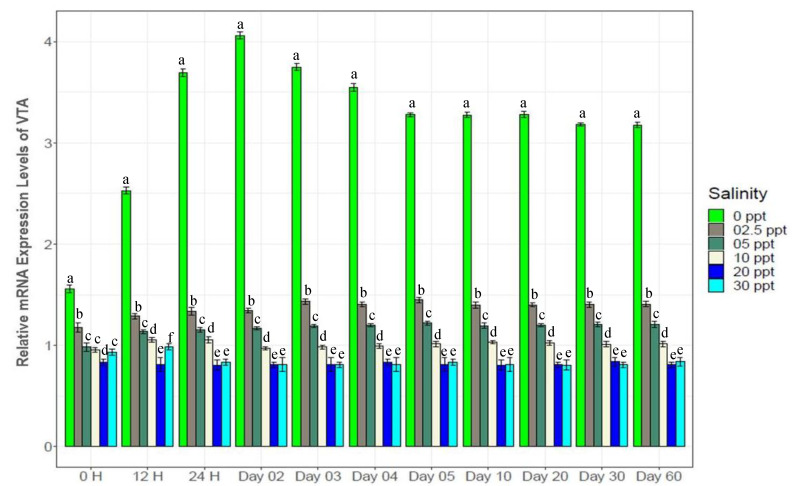
Changes in expression pattern of V-type (H^+^) ATPase (mean ± S.D.) at six different salinity levels. N = 3 individuals per sampling time. Different letters above the bars indicate significant differences among treatments at 5% level of significance.

**Figure 10 biology-10-01220-f010:**
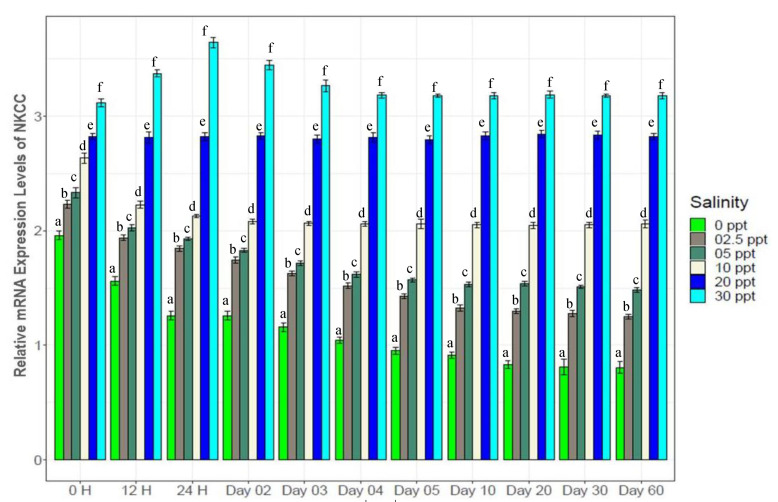
Changes in expression pattern of Na^+^/K^+^/2Cl^−^ Co-transporter (mean ± S.D.; N = 3) at six different experimental salinities. Different letters above the bars indicate significant differences among treatments at 5% level of significance.

**Figure 11 biology-10-01220-f011:**
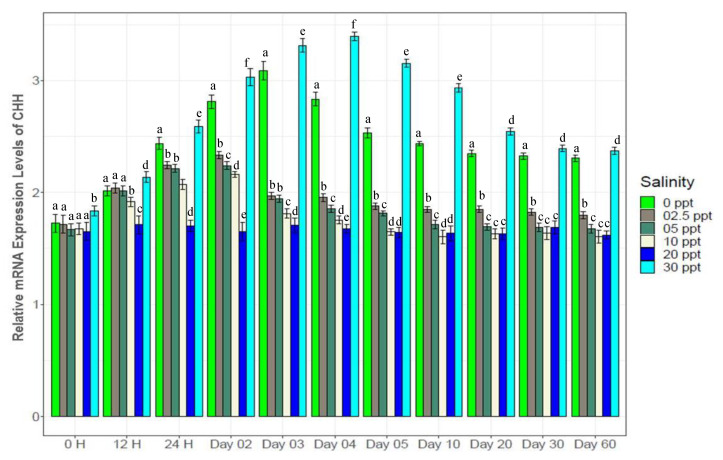
Expression pattern of crustacean hyperglycemic hormone (CHH) (mean ± S.D.; N = 3) at six different salinities. Different letters above the bars indicate significant differences.

**Figure 12 biology-10-01220-f012:**
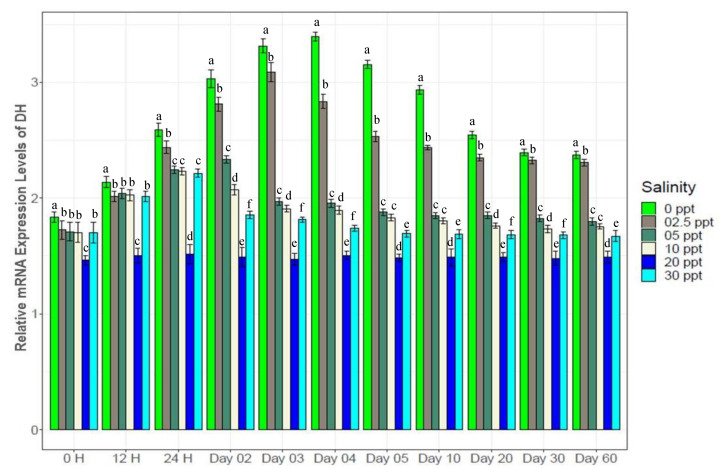
Expression of diuretic hormone (DH) (mean ± S.D.; N = 3) among experimental salinities. Different letters above the bars indicate significant differences among treatments at 5% level of significance.

**Figure 13 biology-10-01220-f013:**
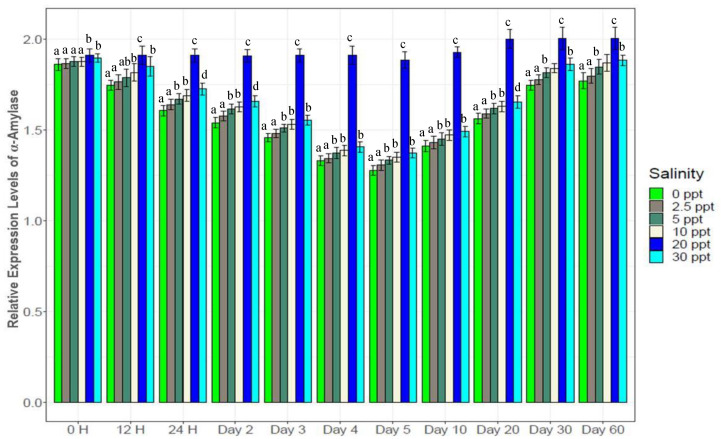
Expression of α-amylase (mean ± S.D.; N = 3) at different salinities. Different letters above the bars indicate significant differences among treatments at 5% level of significance.

**Figure 14 biology-10-01220-f014:**
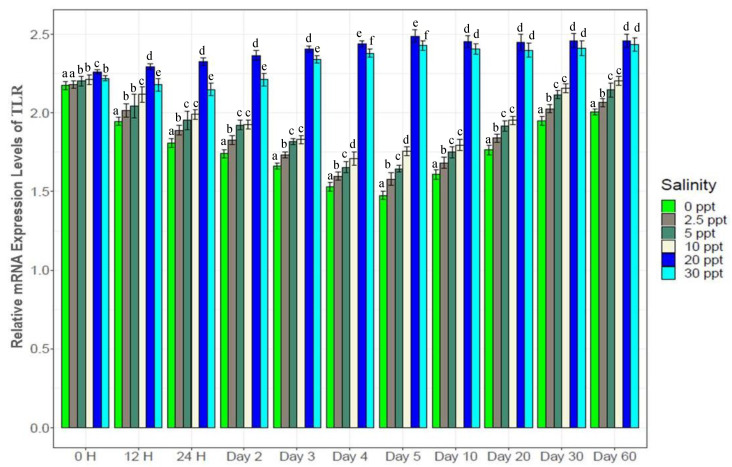
Changes in expression pattern of toll like receptor (TLR) (mean ± S.D.; N = 3) at six different experimental salinities. Different letters above the bars indicate significant differences among salinities at 5% level of significance.

**Figure 15 biology-10-01220-f015:**
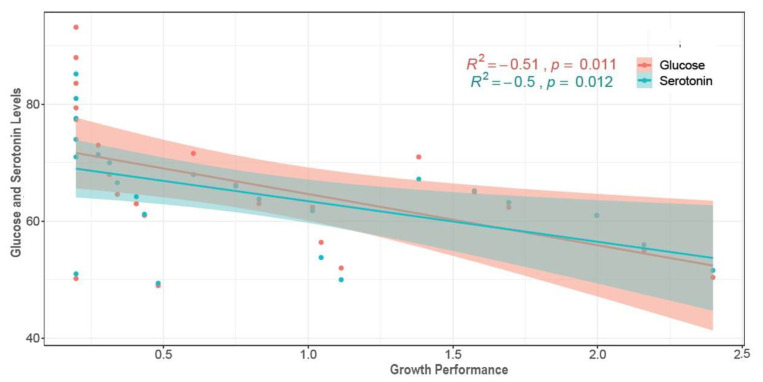
Correlation plot between growth performance (X axis) and hemolymph glucose and serotonin levels (Y axis) for experimental *P. monodon* individuals at six different temperatures.

**Table 1 biology-10-01220-t001:** Primer details for the selected candidate genes for gene expression study.

Gene with ID	Primer Sequence (5′-3′)	Size	Annealing Temperature	Reference
α-Amylase	**F**: TGGACTCGACAACTAGC	118	59 °C	[55]
	**R**: TAGATGCACACTCGCTC			
Crustacean Hyperglycemic Hormone (CHH)	**F**: TTCTGCAGTCTCTCCAACGA	120	59 °C	[56]
	**R**: GCGACGTATGGTTTCCTGTA			
Diuretic Hormone (DH)	**F**: ACAACTCGGCTCTGGTGTTC	110	60 °C	[48]
	**R**: CTCGGCCTAGACCCAAGTC			
Na^+^/H^+^ Exchanger (NHE)	**F**: TTGGAGGAGGGGTCTACCTT	123	59 °C	[48]
	**R**: GCTCTCTCCAAAGACAAGCA			
Na^+^/K^+^-ATPase (NKA)	**F**: CACCCCACCCAAACAAACT	120	60 °C	[55]
	**R**: TCGTGAACTCTTGCTTTCTTGA			
Na^+^/K^+^/2Cl^−^ Co-transporter (NKCC)	**F**: GGGTCACCAGGGTCCAGAT	115	56 °C	[49]
	**R**: TAGCACCAGCAACAATTCCA			
V-type H^+^-ATPase (VTA)	**F**: GTCGATTTTCCGAAGCGAAT	105	61 °C	[55]
	**R**: GCCTTCCAAGTAGCGAAGC			
Toll Like Receptor (TLR)	**F**: CTT AGC CTT GGA GAC AAC	118	53 °C	[26]
	**R**: GAT GCT TAA CAG CTC CTC			
Elongation factor 1-alpha (EF1α)	**F**: TTC CGA CTC CAA GAA CGA CC	122	60 °C	[49]
	**R**: GAG CAG TGT GGC AAT CAA GC			

**Table 2 biology-10-01220-t002:** Salinity specific growth and survival of black tiger shrimp (*Penaeus monodon*). Different superscripts (a–d) across rows indicate significant differences. DWG = daily weight gain, SGR = specific growth rate, FCR = feed conversion ratio.

	0‰	2.5‰	5‰	10‰	20‰	30‰
Initial Weight (BW_i_) (g)	0.21 ^a^ ± 0.04	0.21 ^a^ ± 0.03	0.20 ^a^ ± 0.05	0.20 ^a^ ± 0.04	0.20 ^a^ ± 0.05	0.21 ^a^ ± 0.03
Final Weight (BW_f_) (g)	1.35 ^a^ ± 0.24	1.59 ^b^ ± 0.36	1.73 ^b^ ± 0.46	2.07 ^c^ ± 0.35	2.35 ^d^ ± 0.39	1.98 ^c^ ± 0.34
DWG (%)	8.56 ± 0.66	10.95 ± 0.86	12.06 ± 0.91	15.58 ± 0.92	17.92 ± 0.76	14.05 ± 0.79
SGR (%)	3.1 ± 0.14	3.37 ± 0.26	3.6 ± 0.16	3.89 ± 0.27	4.11 ± 0.36	3.74 ± 0.25
Feed Intake (g g^−1^ day^−1^)	0.162 ± 0.02	0.161 ± 0.03	0.158 ± 0.02	0.158 ± 0.04	0.157 ± 0.01	0.156 ± 0.02
FCR	1.89	1.44	1.31	1.04	0.89	1.11
Survival (%)	61 ^a^	64 ^a^	71 ^b^	76 ^b^	86 ^c^	83 ^c^

**Table 3 biology-10-01220-t003:** Free amino acid (FAA) profiles in the hemolymph (nmol mL^−1^) of black tiger shrimp (*Penaeus monodon*). Different superscripts (a–d) across the rows indicate significant difference. ‘*’ indicates amino acids with particular roles specific for osmoregulation while ‘**’ indicates essential amino acids; EAA = essential amino acids; TFAA = total free amino acid.

Amino Acids	Salinity					
	0‰	2.5‰	5‰	10‰	20‰	30‰
Alanine *	586.3 ± 41.2	561.6 ± 39.6	537.1 ± 40.2	496.5 ± 38.4	406.7 ± 43.8	453.6 ± 33.4
β-Alanine	13.8 ± 1.1	13.1 ± 1.2	11.2 ± 1.1	10.6 ± 0.8	8.76 ± 0.7	9.66 ± 0.9
Arginine **	307.5 ± 3.5	324.8 ± 3.7	353.7 ± 4.6	388.3 ± 4.9	419.8 ± 5.8	428.6 ± 6.2
Asparagine	41.9 ± 2.8	43.7 ± 3.2	46.6 ± 3.5	49.5 ± 4.1	56.2 ± 4.3	54.6 ± 3.9
Aspartic acid	26.4 ± 3.8	26.1 ± 1.7	25.4 ± 2.9	24.6 ± 3.1	23.6 ± 3.6	24.1 ± 2.15
α-AAA	15.1 ± 2.4	16.3 ± 1.7	18.4 ± 1.9	20.7 ± 2.1	22.6 ± 1.9	21.9 ± 1.6
α-ABA	15.4 ± 1.1	10.1 ± 0.9	8.6 ± 0.7	6.6 ± 0.8	4.9 ± 0.4	5.3 ± 0.6
Citrulline	10.4 ± 0.6	12.6 ± 0.5	13.2 ± 0.6	13.9 ± 0.7	15.4 ± 0.3	14.8 ± 0.4
Cystathionine	11.5 ± 2.1	8.93 ± 1.4	6.98 ± 0.8	5.28 ± 0.6	3.97 ± 0.4	4.57 ± 0.5
Cystine	4.73 ± 0.4	4.96 ± 0.5	5.14 ± 0.3	6.11 ± 0.5	6.71 ± 0.4	6.02 ± 0.7
Glutamic acid	62.9 ± 4.4	60.2 ± 4.6	58.9 ± 3.9	56.7 ± 3.3	53.6 ± 3.4	55.4 ± 3.6
Glutamine	196.8 ± 10.4	206.3 ± 31.6	216.5 ± 26.2	226.4 ± 33.4	242.8 ± 39.1	230.4 ± 38.4
Glycine *	396.5 ± 7.6	358.6 ± 9.5	329.6 ± 8.6	356.7 ± 9.8	379.6 ± 6.9	391.5 ± 9.9
Histidine **	44.5 ± 2.9	38.4 ± 2.2	39.4 ± 1.8	37.3 ± 2.1	33.6 ± 1.9	34.8 ± 1.7
Isoleucine **	26.4 ± 1.1	24.1 ± 1.2	22.9 ± 1.5	22.3 ± 1.7	20.3 ± 1.4	21.2 ± 1.3
Leucine **	38.4 ± 1.9	35.7 ± 2.4	34.2 ± 1.7	31.8 ± 1.4	29.7 ± 2.2	30.6 ± 1.6
Lysine **	41.5 ± 2.7	47.5 ± 3.1	51.6 ± 2.8	56.2 ± 3.9	63.9 ± 4.3	63.4 ± 3.8
Methionine **	3.11 ± 0.5	4.38 ± 0.4	4.68 ± 0.5	5.01 ± 0.6	5.91 ± 0.2	6.06 ± 0.7
Ornithine	39.1 ± 3.4	38.4 ± 4.1	35.6 ± 4.1	31.7 ± 3.6	26.38 ± 4.4	22.41 ± 2.9
Phenylalanine **	19.9 ± 1.2	21.3 ± 1.1	24.7 ± 1.7	25.7 ± 1.4	30.6 ± 1.5	29.9 ± 1.8
Proline *	1086 ± 56.7	1068 ± 54.1	1026 ± 43.9	943.5 ± 50.9	930.6 ± 43.8	923.5 ± 60.8
Serine	55.3 ± 4.1	63.2 ± 5.6	72.4 ± 5.9	79.6 ± 6.2	87.8 ± 3.9	86.1 ± 5.4
Taurine	506.8 ± 23.7	486.7 ± 33.2	463.2 ± 32.6	408.5 ± 41.7	384.1 ± 10.4	390.4 ± 39.4
Threonine **	47.2 ± 3.9	47.6 ± 3.6	51.6 ± 3.9	66.4 ± 4.1	86.3 ± 6.5	87.1 ± 5.2
Tryptophan **	3.92 ± 0.8	4.02 ± 0.7	4.58 ± 1.2	5.09 ± 0.9	6.13 ± 0.8	6.27 ± 1.1
Tyrosine	3.98 ± 0.6	4.2 ± 0.4	4.65 ± 0.7	5.1 ± 0.6	6.2 ± 0.7	5.6 ± 0.5
Valine **	31.2 ± 4.26	34.4 ± 3.9	41.7 ± 4.25	47.8 ± 4.38	52.3 ± 4.18	51.6 ± 4.39
∑EAAs	563.63 ^a^ ± 5.3	582.2 ^a^ ± 5.6	629.1 ^b^ ± 8.6	685.9 ^c^ ± 9.3	748.5 ^d^ ± 9.6	759.5 ^d^ ± 6.9
TFAAs	3181.9 ^a^ ± 61	3605.2 ^b^ ± 60	3612.6 ^b^ ± 62	3629.3 ^b^ ± 76	3708.6 ^c^ ± 71	3669.8 ^bc^ ± 66

**Table 4 biology-10-01220-t004:** Free fatty acid (FFA) profiles (% of total fatty acids) in the gill tissue of black tiger shrimp (*P*. *monodon*). Data represent mean ± SE. Different superscripts indicate significant differences among treatments at *p* < 0.05. ∑SFAs = total saturated fatty acids, ∑MUFAs = total mono-unsaturated fatty acids, ∑PUFAs = total poly-unsaturated fatty acids, ∑HUFAs = total highly unsaturated fatty acids.

Fatty Acids	Salinity					
	0‰	2.5‰	5‰	10‰	20‰	30‰
C14:0	0.46 ± 0.05	0.41 ± 0.04	0.38 ± 0.04	0.37 ± 0.03	0.30 ± 0.02	0.35 ± 0.03
C15:0	0.42 ± 0.06	0.37 ± 0.05	0.31 ± 0.05	0.29 ± 0.04	0.27 ± 0.03	0.28 ± 0.02
C16:0	15.54 ± 0.79	14.86 ± 0.76	14.66 ± 0.81	14.34 ± 0.61	13.96 ± 0.62	14.26 ± 0.59
C17:0	0.53 ± 0.04	0.47 ± 0.03	0.46 ± 0.05	0.46 ± 0.04	0.44 ± 0.03	0.45 ± 0.03
C18:0	7.95 ± 0.08	7.81 ± 0.06	7.75 ± 0.06	7.56 ± 0.08	6.86 ± 0.06	7.66 ± 0.07
C20:0	0.98 ± 0.04	0.93 ± 0.03	0.89 ± 0.02	0.88 ± 0.04	0.76 ± 0.03	0.86 ± 0.02
C22:0	0.96 ± 0.06	0.86 ± 0.05	0.78 ± 0.04	0.77 ± 0.03	0.74 ± 0.05	0.71 ± 0.04
∑SFAs	26.86 ^a^ ± 0.66	25.71 ^ab^ ± 0.69	25.23 ^b^ ± 0.64	24.67 ^bc^ ± 0.61	23.33 ^c^ ± 0.56	24.57 ^bc^ ± 0.65
C16:1n7	3.46 ± 0.05	3.26 ± 0.06	3.11 ± 0.09	2.93 ± 0.07	2.76 ± 0.06	2.89 ± 0.08
C18:1n9	19.83 ± 0.18	19.46 ± 0.19	19.31 ± 0.21	19.06 ± 0.16	18.61 ± 0.11	18.99 ± 0.13
C18:1n7	3.78 ± 0.09	3.62 ± 0.06	3.49 ± 0.07	3.36 ± 0.05	2.98 ± 0.02	3.18 ± 0.06
C20:1n7	1.04 ± 0.04	0.99 ± 0.05	0.96 ± 0.03	0.93 ± 0.04	0.83 ± 0.01	0.89 ± 0.02
∑MUFAs	28.11 ^a^ ± 0.32	27.33 ^ab^ ± 0.31	26.87 ^ab^ ± 0.33	26.26 ^bc^ ± 0.29	25.18 ^c^ ± 0.22	25.95 ^bc^ ± 0.28
C18:2n6	7.86 ± 0.14	7.69 ± 0.09	7.66 ± 0.13	7.44 ± 0.11	6.96 ± 0.06	7.33 ± 0.09
C18:3n3	0.87 ± 0.05	0.82 ± 0.06	0.74 ± 0.02	0.71 ± 0.04	0.59 ± 0.03	0.64 ± 0.01
C20:2n6	2.45 ± 0.05	2.41 ± 0.07	2.39 ± 0.10	2.34 ± 0.06	2.26 ± 0.02	2.29 ± 0.08
C20:3n6	0.28 ± 0.05	0.26 ± 0.03	0.25 ± 0.04	0.22 ± 0.02	0.19 ± 0.01	0.20 ± 0.03
C20:4n6	17.03 ± 0.21	16.96 ± 0.23	16.76 ± 0.14	16.50 ± 0.09	16.02 ± 0.15	16.46 ± 0.17
C20:5n3	10.42 ± 0.23	10.34 ± 0.27	10.23 ± 0.21	9.97 ± 0.16	9.68 ± 0.18	9.86 ± 0.12
C22:6n3	5.41 ± 0.08	5.46 ± 0.07	5.56 ± 0.09	5.76 ± 0.08	5.96 ± 0.04	6.35 ± 0.22
∑PUFAs	44.32 ^a^ ± 0.17	43.94 ^a^ ± 0.24	43.59 ^ab^ ± 0.28	42.94 ^b^ ± 0.26	41.66 ^c^ ± 0.12	43.12 ^b^ ± 0.24
∑n-3PUFAs	15.56 ^a^ ± 0.12	15.36 ^a^ ± 0.13	15.24 ^ab^ ± 0.14	15.06 ^ab^ ± 0.10	14.35 ^b^ ± 0.11	14.98 ^ab^ ± 0.09
∑n-6PUFAs	24.52 ^a^ ± 0.10	24.31 ^a^ ± 0.11	24.16 ^ab^ ± 0.13	23.96 ^ab^ ± 0.09	23.32 ^b^ ± 0.07	23.76 ^ab^ ± 0.08
n-3/n-6	0.64 ± 0.03	0.63 ± 0.02	0.63 ± 0.02	0.62 ± 0.04	0.61 ± 0.01	0.62 ± 0.03
∑HUFAs	35.06 ^a^ ± 0.25	34.76 ^ab^ ± 0.37	34.56 ^ab^ ± 0.34	34.11 ^b^ ± 0.27	33.60 ^b^ ± 0.32	34.06 ^b^ ± 0.29

## Data Availability

We do not have any data to share. All of the data generated have been used and presented as tabular and graphical forms in this manuscript.

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
