# Peer review of "Effects of Salinity on Physiological, Biochemical and Gene Expression Parameters of Black Tiger Shrimp (Penaeus monodon): Potential for Farming in Low-Salinity Environments"

_biology, 2021, doi:10.3390/biology10121220_

Round 1

Reviewer 1 Report

The authors wanted to show the effect of salinity of the water in the survival and growth of crustacean, especially by analyzing the effects of the salinity on gene transcritption, biochemical, physiological and immunological factors.

Knowing these effects is of importance for the farming of crustacean, however the authors should improve at least the manuscript to understanding the exact goal of studying these parameters and especially what are the conclusions and analisis they can make from these results.

In general, I find the manuscript confusing. I would summarize the abstract and introduction to try to make things more clear, not go some much in some details, and reorganize the introduction.

figures should be modified in general (scale) and title and description of the figures should be improved. a little bit more description, statistics...

Reviewer 2 Report

     Black tiger shrimp comes inshore to spawn and young life stages remain in brackish water for a time before moving to sea. Culture of the species is best done at 20 ppt, but what would be the result in water of lesser salinity? Rahi et al. looked at a variety of metrics, including survival, growth, physiological, biochemical, and gene expression, and thereby achieved a more holistic understanding of the effects of different salinities upon the species. The work is basically sound, though the manuscript will need considerable work before it proves ready for acceptance. The prose is rough – I have marked the manuscript to guide its revision, and hope the authors take this as constructive. The authors must not oversell their results as indicating that the species can be raised (profitably) in freshwater. They did not really test that, certainly not under commercial production conditions. Other more context-dependent comments follow.   

     Abstract. – At line 17 and throughout the manuscript, “survivability” should be replaced with “survival”.

     Methods. – At line 150, is the cited feeding rate per day? It would seem so, but should be made explicit.

     At line 174, Qubit is in Ontario in Canada, but more generally, the locations of manufacturers of all products mentioned should include mention of the city where located.

    At line 275 and more generally throughout the document, amylase is presented as a “growth gene”. Amylase is an enzyme that breaks down carbohydrates – to call it a growth gene is misleading. That is, it is not a growth-promoting factor like growth hormone. Best to call it a digestive enzyme.

     At lines 285-286, it is not enough to say that relative gene expression values were calculated in R, that’s an operating system. The reader should be told what computational package was used.

     The supporting citation for Jolliffe and Cadima (2016) cited at line 287 is not presented in the References section. 

     Results. – In Table 2, the metrics in the first column should be spelled out so that the table can be understood independently of the text.

     Figure 1 is redundant with Table 2, and can be deleted.

     The authors applied salinity changes to the respective groups of shrimp. Once applied, the salinities were stable, and did not fluctuate. Fluctuation is ongoing change, as for example, salinity at a point in an estuary rises and falls with the tides. At line 317 and other places in the manuscript, the authors must write of salinity changes, not fluctuations.

     The captions for figures 7-12 and 14 should spell out the names of the genes at issue so the reader can understand each one independently of the text. Additionally, the sizes of the figures and fonts vary, and some seem too large.

     Discussion. – At line 458, I’d expect that higher O2 consumption would lead to slower growth. Indeed, that expectation is communicated at line 462, an so I’d ask for revision at line 458. Additionally, do the Loughland and Seebacher and Rahi et al. citations support the statement made at line 458? The papers have titles that do not seem directly related to the assertion made.

     At line 491, the ability of the shrimp to change their osmolality levels underlies their ability not only to disperse between brackish and marine environments as written, but also to adapt to tidally driven changes in salinity at any one location.

     At line 543, glucose and serotonin are not “important hurdles to growth”, it is stress that impedes growth. Heightened glucose and serotonin levels are part of the shrimp’s stress response.

     At line 576, high expression of amylase CONTRIBUTED to superior growth performance. As written, the sentence oversells the importance of this gene.

     At line 601, findings of this study SUGGEST that with proper acclimation, the shrimp can be farmed in low salinity or freshwater conditions. The findings should not be oversold. The authors did not demonstrate profitable shrimp production in freshwater. This was a study of a variety of markers in a laboratory.

     At line 607, the sentence is quite simply indefensible. Good farming practice cannot render low-salinity environment pathogen-free. The pathogens are ubiquitous. Good farming practice might minimize outbreaks of disease.   

     References. – The font of the references is larger than for the rest of the manuscript. I have marked departures from journal citation stylistics. A url should be given for FAO (2019). The Jolliffe and Cadima citation is not given.
